# Extending a Conceptual Framework for Junior Doctors’ Career Decision Making and Rural Careers: Explorers versus Planners and Finding the ‘Right Fit’

**DOI:** 10.3390/ijerph17041352

**Published:** 2020-02-20

**Authors:** Beatriz Cuesta-Briand, Mathew Coleman, Rebekah Ledingham, Sarah Moore, Helen Wright, David Oldham, Denese Playford

**Affiliations:** 1Rural Clinical School of Western Australia, Faculty of Health and Medical Sciences, University of Western Australia, 6280 West Busselton, Australia; beatriz.cuestabriand@rcswa.edu.au (B.C.-B.); mathew.coleman@rcswa.edu.au (M.C.); bek.ledingham@rcswa.edu.au (R.L.); helen.wright@rcswa.edu.au (H.W.); denese.playford@rcswa.edu.au (D.P.); 2Western Australia Country Health Service, 6000 Perth, Australia; david.oldham@health.wa.gov.au

**Keywords:** junior doctor, rural workforce, career decision making

## Abstract

This study uses data from a Rural Clinical School of Western Australia (RCSWA) and WA Country Health (WACHS) study on rural work intentions among junior doctors to explore their internal decision-making processes and gain a better understanding of how junior doctors make decisions along their career pathway. This was a qualitative study involving junior doctor participants in postgraduate years (PGY) 1 to 5 undergoing training in Western Australia (WA). Data was collected through semi-structured telephone interviews. Two main themes were identified: career decision-making as an on-going process; and early career doctors’ internal decision-making process, which fell broadly into two groups (‘explorers’ and ‘planners’). Both groups of junior doctors require ongoing personalised career advice, training pathways, and career development opportunities that best “fit” their internal decision-making processes for the purposes of enhancing rural workforce outcomes.

## 1. Introduction

Developing a sustainable rural health workforce to ensure equitable access to healthcare professionals and services for rural patients is an international challenge [1]. In Australia, the Rural Clinical Schools (RCSs) program has had positive workforce outcomes [2], and pre-vocational rural clinical placements have been shown to have a positive effect on rural practice intention [3]. However, interest in rural practice in the early post-graduate years is not fully reflected in actual practice location [4,5], and a number of barriers to rural practice have been identified, including a lack of rural training programs and personal and family reasons [6].

Most of the Australian evidence on junior doctors’ career choices, consists of quantitative data on current practice location [4,7,8,9,10], and qualitative studies largely focused on the impact of rural training placement [3,11,12] or the identification of factors influencing career decision making [5,6,13]. To our knowledge, no evidence has been published on the exploration of junior doctors’ internal decision-making processes, and there have been calls for more qualitative research to fully understand how medical graduates make choices along the different pathways to rural practice [14].

Pfarrwaller and colleagues [15] developed a conceptual framework of medical students’ primary care career choice. The model consists of two parts: a central part representing students’ internal processing of career choice pathways, and an outer part representing the different systems of influence on career decision making. We have described the testing of the outer part of the model in our first paper in this Special Edition on Rural and Remote Health [16].

The central or inner part of the model broadly draws from social cognitive theory, as applied to career choice [17], whilst the outer part draws from ecological theory [18]. The central part of the model focuses on the decision-making process and is divided into three components: personal characteristics; the decision process (during which students match their personal interests with their perceived characteristics of different specialties) [17]; and career choice [15] (See Figure 1). In this paper, we use data from a Rural Clinical School of Western Australia (RCSWA) study on rural work intention among junior doctors to explore the internal decision-making process and test the inner part of the model produced by Pfarrwaller and colleagues [18]. A better understanding of how junior doctors make decisions along their career pathway will help inform innovative solutions and policy settings to develop better outcomes in developing and retaining a sustainable rural medical workforce.

## 2. Material and Methods

This was a qualitative study informed by the principles of phenomenology insofar as it was interested in participants’ lived experiences. Ethics approval was obtained by the WA Country Health Service (WACHS) Human Research Ethics Committee (HREC) in November 2018 (approval number 1130).

Participants were recruited among junior doctors in postgraduate years (PGY) 1 to 5 undergoing training in Western Australia (WA) who participated in an online survey exploring the factors influencing the decision to pursue rural work administered in October 2018 and September 2019. A link to the survey was distributed to all postgraduate medical staff of the three Primary Employer Health Services in WA and the WA Country Health Service (WACHS) through their Medical Education Officer using staff directories. A follow-up email was distributed two weeks later. Survey respondents were asked to include their contact details if they wished to be contacted for a follow-up interview, and those who responded to a follow-up email were invited to participate. A purposive sampling approach was adopted to ensure a broad representation of experiences.

The data was collected through semi-structured telephone interviews with an average duration of 33 min. The interview schedule covered four broad topics (work since graduation; career intentions; training and support; and future career plans) and consisted of general open-ended questions and prompts to explore specific aspects within each topic. The semi-structured format ensured the consistent exploration of all topics across all interviews whilst allowing for the introduction of new themes.

The process of data collection and analysis was iterative. All interviews were transcribed, and the resulting transcripts were imported into NVivo 12 [19] and subjected to thematic analysis. The data analysis was mainly inductive and followed the four steps described by Green and colleagues: immersion in the data; coding; creating categories; and identifying themes [20]. Transcripts were read and coded separately by the research team, and a list of codes was developed and refined as coding progressed [21]. The conceptual framework by Pfarrwaller and colleagues—specifically the central part of the model [15]—was used to guide the interpretation of the data, the research team agreed on the main themes. Rigour was enhanced through investigator triangulation, through team member checking, coding validation, peer debriefing [22], and the use of NVivo [23].

## 3. Results

### 3.1. Sample

A total of 21 participants were interviewed; their characteristics are shown in Table 1.

### 3.2. Themes

Two main themes were identified: (1) career decision making as an on-going process and (2) exploration versus planning in career-decision making. These themes are discussed below and are illustrated with contextualised quotes. Sites have been omitted to maintain confidentiality.

#### Career Decision Making as an On-Going Process

Overall, career decision making emerged as a dynamic and continuous process. Most doctors in our study were still making career decisions regarding speciality or future practice location at the time of the interview, and change featured strongly in their descriptions of their careers. There was evidence of change of career intention from medical school among the majority of our junior doctors, and this was strongly influenced by exposure to different specialties and to rural practice.

### 3.3. Change of Intention, Change of Location

The comparison of reported past and current intended speciality and practice location showed a significant degree of change, especially relating to speciality intention. As shown in Table 2, 11 participants reported having changed speciality intention since medical school (in bold), whilst six were still undecided (in italics). It is also noteworthy that only six participants had been admitted into vocational training programs (underlined).

Although there was less evidence of a change of intention with regards to rural or urban practice location, eight doctors had changed their original intention and two were still undecided. For those who reported having a rural intention during medical school and a current urban location intention, the change appeared to be unrelated to their rural background status and tended to be associated with perceived lack of rural pathways for their speciality of choice or with perceived greater career opportunities for their partners in urban settings.

There was also evidence that some junior doctors pro-actively sought relocation during their pre-vocational training years in order to progress their career. One urban-based doctor with strong rural intention reported having successfully applied for a one-year secondment from a city hospital to a remote location in order to gain rural exposure, whilst another had sought a six-month contract with a regional hospital in order to gain Emergency Department exposure. Rural-based doctors also sought to move between regional locations to address perceived training gaps relating to case exposure, especially with regards to Aboriginal health. One doctor explains:
[Southern location], the majority of your patients are Caucasian, over 60 and presenting with exact stages of chronic illness. Whereas in [northern location], the majority of the patients are Indigenous, they’re usually somewhere between 40 and 60 but often younger, and they’re usually presenting with more acute illness that has not been diagnosed, and it’s just different. It’s a different population to work with and then, on top of that, […] it’s also a different group of doctors who have a different philosophy on practice and a different sort of mindset to practice. […] They’re constantly doing the same job over and over again in [southern location].*(I04; Male; FGY3)* 

### 3.4. The Role of Exposure

The narratives of the junior doctors in our study showed that exposure to speciality and rural practice influenced career decision making, solidifying or reshaping professional interests through a process of reassessment of how what they experienced or were witnesses to matched their preferences and perceived attributes.

There was evidence that positive experiences during medical school had a strong impact on career choices. RCSs were widely perceived as a positive formative experience, and the description of an RCS year often elicited emotive responses from doctors who had been part of the program (‘fantastic’, ‘great’, ‘the best thing’). Furthermore, RCSs were widely reported as triggering an interest in rural practice, which many were still pursuing. Other positive experiences during medical school had a similar impact on career decision making, as this doctor—with a rural background but no RCS experience—explains:
So I’d gone through [medical school] thinking I would always live and work in the city till after I did my rural placement. My favourite areas throughout medicine, I’ve absolutely loved GP placements, and I loved emergency, and I really, really did love obstetrics as well. And in my final year I started to think the areas of work where I can combine those three things are working in rural and remote areas in Australia. Then I did my final year rural placement in a smallish town with a small hospital where I was placed with somebody who did exactly that, he was a GP who covered emergency and did obstetrics and some minor surgical stuff as well. And I was sold that that was definitely what I wanted to do from then.*(I03; Female; PGY2)* 

There was also evidence that direct experience or witnessing doctors working in specific specialties during early pre-vocational years also had an impact on career decision making. In some cases, this exposure impacted negatively on earlier career intentions, as this doctor—who initially intended to pursue a rural GP career—reveals when she describes her experience of witnessing the work of rural GPs:
They seemed to have a very good lifestyle, but hearing about the work that they do, they seem to be quite overworked at lot of the time, and they looked really stressed. So I thought maybe it wasn’t the path for me.*(I02; Female; PGY3)* 

The above quote reveals a preference for a different work–life mix, which they had observed and was consistent with the narratives of many doctors in our sample for whom lifestyle factors played an important part in career decision making. There was evidence that doctors assessed their experiences or observations against their own personality, and whilst there were few references to personal attributes, doctors in our sample often talked about personal preferences or interests (for example, a preference for continuity of care was expressed by most junior doctors with a general practice intention) influencing their career choices. Reflecting on a preference for ‘working quickly’, an urban-based doctor who initially was interested in becoming a physician commented:
Once I started working I found that working in Emergency suited my personality better. […] I guess I like the variety, I like working quickly. I actually preferred shift work where you didn’t have to think about things when you left the hospital. […] I don’t think you can walk away from everything. But when I looked at what consultant physicians do and consultant emergency doctors do, I thought that emergency suits my personality better. And in the long-term it’s probably what I would like to do. It fits in with what my lifestyle is like. *(I11; Female; PGY2)* 

#### Exploration versus Planning

The narratives of our doctors demonstrated different experiences of and attitudes towards career planning. Independent of their speciality and rurality intention, doctors appeared to fall into two distinct categories in the analysis: ‘explorers’ or ‘planners’.

### 3.5. The Explorers

Eleven doctors in our sample appeared to be at the early stages of career planning, insofar as they were still exploring their career options. Whilst career indecision was expected and indeed identified among PGY1 doctors (Interns), there was also evidence of indecision among doctors in later pre-vocational training years. Two PGY3 doctors reported being still undecided about their career speciality, and two more demonstrated a certain degree of hesitancy despite having a more defined speciality intention, as this quote reflects:
I’m almost sure that I want to end up doing rural psychiatry, but I want to have a bit of a taste before I apply for the program.*(I02; Female; PGY3)* 

The above quote encapsulates the process of the elimination (‘ruling things out’) approach to career decision making, which characterised the ‘explorers’ in our sample. Doctors spoke of trying different specialties and settings (hospital/community, urban/rural) before they decided which fitted their personality type and preferences better. One doctor with a rural background, a strong rural intention, and an inclination for rural GP practice explained how he had kept an ‘open mind’ with regards to his career during medical school, later adding:
I still am keeping an open mind but I’m kind of removing things that I don’t enjoy as much. […] So, I guess at this point I want to be a generalist. I like a bit of everything, it keeps it interesting, it keeps it fresh and I have an interest in anaesthetics because I did an elective in [remote location] under the Royal Flying Doctors. There’s a heavy anaesthetic kind of rotation there and that really interests me. So, at the moment, in a combination of keeping things broad and a bit of everything, so maybe a generalist with an interest in anaesthetics, so remote rural GP with anaesthetics as a speciality.*(I06; Male; PGY2)* 

Similarly, another doctor explained how she had come to the realisation that general practice might be the right speciality for her, although her narrative suggests a certain ambivalence and unreadiness to commit. She initially spoke about enjoying women’s health and Aboriginal health in medical school, adding:
Those two areas were where I wanted to go. And I guess that all sort of leads me towards general practice. But yeah, I’ve sort of been unsure most of medical school and not really wanting to say to people that I do want to do general practice. I’ve been sort of wanting to find out what else is out there. But now that I’m in the hospital system, I do think general practice would probably be right for me.*(I07; Female; PGY2)* 

With the exception of two doctors, whose narratives demonstrated a higher degree of goal setting despite still being uncommitted, there was limited evidence of proactive career planning among the ‘explorers’. Instead, doctors in this category tended to demonstrate a somewhat passive approach, characterised by fortuitous rather than deliberate decision-making, and a preference for allowing career and life events to unfold naturally, as this quote illustrates:
It doesn’t matter where I really see myself in five years’ time, the likelihood of that happening is probably non-existent, so I just kind of take it as it comes. […] I don’t focus on where I’ll end up being. I’ve never said to myself, “I want to be a consultant in this field, doing this, in this place, in five years’ time”, because at the end of the day the likelihood is if I say that now I probably actually won’t want to be that when I get to that point. And so I figured that I’ll fall into something, somewhat naturally.*(I04; Male; PGY3)* 

Similarly, an urban-based doctor described his approach:
I sort of gave myself three years to decide what you would like to do and go with the flow with the rotations you were given and then ultimately I am now coming to a conclusion that community medicine or GP with some interest area or speciality area is probably what I will be heading down.*(I17; Male; PGY3)* 

Later in the interview, the same doctor, who was yet to commit to a speciality, commented:
Once I have made that decision then you become more engrossed in learning the specific things for that career, as opposed to what I have probably been doing for the last three years, which is try and learn as much as you can from the rotations you have with no real aim of that career pathway, more of just building your general knowledge.*(I17; Male; PGY3)* 

The above quote highlights that although doctors who adopt a more laidback approach to career planning might be at a higher risk of losing focus and direction in their careers, they can also benefit from an extended time being exposed to different specialties and building their general clinical skills.

### 3.6. The Planners

At the other extreme from the ‘explorers’ were doctors whose narratives showed a high level of career planning, demonstrated by a proactive approach to career decision-making, which included the early identification of career goals and the undertaking of purposeful, intention actions to achieve those goals.

Doctors in this category tended to be less likely to report having changed their career intention, and their narratives showed more evidence of proactive decision making from the outset, as demonstrated through their choice of internship hospital. In contrast with the experience of the ‘explorers’, who tended to report making choices about the site of their internship based on personal relationships or familiarity with the hospital, the ‘planners’ tended to report having chosen a specific site based on their career goals. A PGY4 doctor, who, at the time of the interview was in his second paediatrics training year, explained that he had selected the location partly because the hospital had an emergency department that accepted paediatric presentations. Another doctor, who, at the time of the interview was in her first year of rural GP training, explains:
I did my internship at [urban hospital]. I purposefully did not apply for any rural terms because I knew that I wanted to go back rural, and I wanted to get the most city experience that I could in the years that I had to stay in the city.*(I05; Female; PGY4)* 

Unsurprisingly, the ‘planners’ included all those who had already been accepted into vocational training programs. However, they also included doctors in their internship year; one of whom explains why he chose a specific urban hospital:
One, because they have a very large plastic surgery contingent. And two, because they’re also the specialists as far as anaesthetics are concerned. They’re the guns. So it’s nice to be around the people who are at the top end.*(I19; Male; PGY1)* 

As their training progressed, ‘planners’ were more likely to report having undertaken specific actions to achieve their career goals, for example, finding mentors in their intended speciality field, cultivating professional networks, seeking advice from College Board members, approaching specific departments to gain case exposure, and proactively seeking information on specific training pathways. The narratives of the ‘planners’ were also more likely to demonstrate higher levels of self-efficacy and self-advocacy, especially with regards to navigating the complexities of contractual arrangements under which the WA health system operates to progress their career. A doctor with strong rural intention, who started his training in the city and was due to start rural GP training in 2020, explains:
So I’d always planned to do a six-month ED and to do a Certificate and I was planning to do that in 2019. So it was always in the back of my mind. But then [urban tertiary hospital] rotations weren’t finalised and they gave me ‘to be confirmed’ terms for the second half of the year. And then this job came up in [rural location] which a friend who worked down here put me onto. And so it was just the better decision career-wise to shift for that because I knew what I’d be getting out of it.*(I08; Male; PGY3)* 

Similarly, another doctor explained how she had successfully negotiated a service registrar position at a remote hospital to suit her career needs:
I initially applied for the resident medical officer position and I got that. So I got a position as a resident. However, the registrar position opens later in the year, and so I held a resident position for a while until I knew that I had a registrar position. So the registrar position, on top of all this, they usually only give you a six-month contract in a rural location. And so I called and I emailed and I got in touch with both WACHS and the city where I seconded from to express my interest in what I wanted. I wanted to be in [rural location] for 12 months, whether that was a six-month reg, six-month RMO, or ideally a 12-month registrar position. So that took a little bit of orchestration.*(I12; Female; PG5)* 

## 4. Discussion

Results from our study have demonstrated that career decision making among junior doctors is a dynamic, on-going process characterised by a significant amount of change in the initial post-graduation training years. Our results also show that junior doctors have different attitudes towards career planning, which significantly influence their internal decision-making process, and inform how, why, and when they make decisions about speciality and practice location along their individual career pathways.

Our results draw from social cognitive theory and augment previous published models of medical career decision making. Stagg and colleagues proposed a ‘four quadrant model’ (‘the true believers’; ‘the convertibles’; ‘the frustrated’; and ‘the metro docs’) to analyse survey data on career choice and practice location [5]. Although useful to better understand how some external factors might operate, the model does not address internal drivers. Our proposed typology of ‘ explorers’ and ‘planners’ has practical implications insofar as it allows for an understanding of the internal decision-making process governing the direction and choice of career paths with respect to rural location and training pathways using the principles of social cognitive career theory included in the central or inner part of the model by Pfarrwaller and colleagues [15] (See Figure 1). Our results contribute to social cognitive career theory by illuminating how the three components of the inner part of the model operate, especially the ‘decision process’ component and its intention/action/performance/interest cycle [15]. 

With regards to personal characteristics, results from our study show that perceived personality type influenced career decision making, insofar as it helped junior doctors assess whether certain specialties suited them, lending support to the role of experience used to confirm or disconfirm speciality choice [13]. Of note in this study was that rural background featured prominently in the narratives of doctors who grew up in the country, however, rural experience was present in both ‘planners’ and ‘explorers’. Furthermore, neither sex nor RCS participation appeared to be associated with either decision-making type. More research is warranted to further explore how RCS experience, and other pre-graduation exposure and work experience influences career decision making.

In our study, regardless of their level of self-efficacy relating to their professional performance, unsurprisingly, ‘planners’ tended to demonstrate higher levels of self-efficacy at making career decisions (career decision-making self-efficacy) [24] compared with ‘explorers’. Our results also suggest that career decision making self-efficacy (which is a part of ‘process efficacy’ or the perceived ability to manage generic tasks necessary for career planning) [25] may operate independently from professional self-efficacy. These findings have implications for workforce development as ‘planners’ and ‘explorers’ will have different needs regarding career planning.

Our results suggest that ‘explorers’ are more likely to have an external locus of control and therefore ecological factors may play a larger influence on their career decisions. Whilst some ‘explorers’ may be comfortable letting career and life decisions unfold naturally, and may apparently not need much support, others may benefit from guidance so that a lack of defined career path does not result in a feeling of loss or a lack of direction. This has significant implications for rural workforce development, especially for rural background ‘explorers’ with rural intention.

Our findings also highlight the critical role played by exposure to, and opportunities in, speciality and practice location, as junior doctors continuously assess and reassess their experiences and observations against their initial intentions. Whilst it has been argued that medical training in Australia is too long and that doctors should be streamed into speciality training earlier [26], data from our study suggests that although this might be an effective strategy for ‘planners’, committing early to a speciality could be detrimental to ‘explorers’, depriving them of time and opportunity to experience different disciplines and practice locations. Based on these results, we hypothesise that ‘explorers’ will require more intention/action/performance/interest assessment cycles before they are ready to make a career choice, and they will need support and a variety of training opportunities along that pathway. In contrast, ‘planners’ with a specific career goal might need guidance to identify available training pathways that will allow them to achieve their goal.

So, how can early career doctors be supported and afforded opportunities to find the ‘right fit’ for their career development? Furthermore, if junior doctors are considering a rural career, what may influence ‘explorers’ towards a rural direction and what will confirm ‘planners’ to a rural pathway?

Our results support the need for personalised career advice for junior doctors that is responsive and tailored to their personal, family, and social circumstances, and takes into consideration their personal characteristics, their career and personal goal and aspirations, and their career decision-making profile. ‘Planners’ may benefit from up to date, detailed information about how to achieve a career goals in rural practice within formalised training pathways. ‘Explorers’ will benefit from good quality exposure to rural medicine at multiple points in their early career. Existing opportunities such as longitudinal clerkships available in RCS and the John Flynn Placement Program [27] during their university training have been a proven success, and a focus on pre-vocational placements in rural medicine may assist the ‘explorer’ to follow a rural pathway and career path [28].

More broadly, within both generalist and specialist training pathways, uncertainties in the employment, training, and education landscapes (e.g., supervisor availability, training network or program providers, workforce skill-mix demands, etc.) are often magnified in rural settings, providing additional challenges for both ‘planners’ and ‘explorers’ and the way in which they respond to career decision making. These external factors will have differing effects on the internal decision-making processes of ‘planners’ versus ‘explorers’. Medical Educators and Directors of Training will need to contextualise these external factors when supporting junior doctors depending upon their typology for internal decision making. Finally, rural health services, educators and training organisations may need to adapt their career and training pathways to the needs and preferences of junior doctors. This study’s findings suggest that dynamic, continuous processes, using training rotation, annual appraisals, and mentoring schemes assessing how junior doctors are progressing through their training journey, may assist in ensuring the right training and career pathway “fit” for rural intentioned junior doctors.

Overall, our typology of ‘planners’ and ‘explorers’ enhances Stagg and colleagues’ ‘four quadrant model’, as understanding and addressing the needs of ‘explorers’ versus ‘planners’ may be useful to support all four categories of doctors, but especially the ‘convertibles’ and the ‘frustrated’, which has significant implications for rural health workforce planning. Further research is warranted to explore how time (another component of Pfarwaller’s model [15]) influences career decision making as junior doctors navigate their initial training years.

Our study has a number of strengths and limitations. In combination with our first paper [16], we successfully tested the two main components of the model by Pfarwaller and colleagues [15], providing a comprehensive explanation of the internal career decision-making process and how different systems of influence operate and demonstrating that the model can be applied to junior doctors’ career decision making. We included a broad range of junior doctors regardless of practice location, speciality intention, and rural exposure. Furthermore, we explored career decision making ‘in real time’ rather than retrospectively and identified two phenotypically different groups of students regarding their career decision making. We acknowledge some limitations. Firstly, participants self-selected and, given our recruitment strategy, our sample included an overrepresentation of junior doctors with an interest in rural practice; this, however, does not undermine the validity of our results. Secondly, our study focused on one Australian state with a distinctive population distribution and geographic characteristics, and, as a result, some of our findings might have limited applicability in other settings. Finally, our findings need to be understood in the context of the Australian medical training model and will have greater applicability in countries with a similar training model, such as Canada.

## 5. Conclusions

This study broadly identified early career doctors’ internal decision-making process into two groups: ‘explorers’ and ‘planners’. However, career and training decision making is a continuous and dynamic process. The needs, supports, opportunities, and influences on rural training and career decision making for both groups require reflective consideration by medical educators, trainers, and rural health services. Both groups of junior doctors require regular personalised career advice, adaptive training pathways, and opportunities that best “fit” their internal decision-making processes for the purposes of enhancing rural workforce outcomes.

## Figures and Tables

**Figure 1 ijerph-17-01352-f001:**
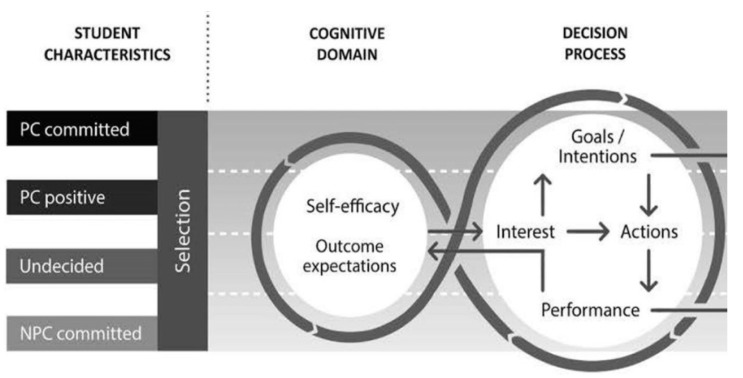
Pfarrwaller Framework-Inner Part (partial reproduction of a larger figure). This figure is a detailed representation of students’ career choice process. Pre-exisiting personal characteristics influence students’ initial interest in primary care; based on this interest, students fit into one of four distinct groups: primary care (PC) committed, primary care positive, undecided and non-primary care (NPC) committed. During this process, students may change career paths based on factors within the cognitive domain and the decision-making process [15].

**Table 1 ijerph-17-01352-t001:** Interview participant—main sample characteristics.

Characteristic	Sub-Group	Total	Total
Sex	Female		11
	Male		10
		*Total*	*21*
PG Year (at interview)	PGY1 PGY2		4 4
	PGY3		8
	PGY4		3
	PGY5		2
		*Total*	*21*
Rural Background	Yes		11
	No		10
		*Total*	*21*
RCS Participation	Yes		14
	No		7
		*Total*	*21*
Specialist Intention	GP		7
	Non-GP		8
	Unsure		6
		*Total*	*21*
Age Group	25-30		17
	31-40		4
		*Total*	*21*

PGY: Postgraduate Year. RCS: Rural Clinical School. GP: General Practitioner.

**Table 2 ijerph-17-01352-t002:** Speciality and practice location intention in medical school and current.

Interview	PGY	Intended Speciality (MS)	Intended Speciality (Current)	Intended Practice Location (MS)	Intended Practice Location (Current)
1	PGY3	GP	**EM & ICM**	Rural (somewhat)	**Urban**
2	PGY3	GP	**Psychiatry**	Rural	Rural
3	PGY2	Obs-Gyn	**Rural GP**	Urban	**Rural**
4	PGY3	Unsure	Unsure	Rural	Rural
5	PGY4	Rural GP	Rural GP	Rural	Rural
6	PGY2	Unsure	**Rural GP**	Rural	Rural
7	PGY2	Unsure	Unsure	Rural	**Unsure**
8	PGY3	Rural GP	Rural GP	Rural	Rural
9	PGY3	Physician	**Rural GP**	Unsure	**Rural**
10	PGY4	-	Psychiatry	Rural	Rural
11	PGY2	Physician	**EM**	Rural	**Urban**
12	PGY5	Neurology	**Surgery**	Rural (somewhat)	Rural
13	PGY5	EM	**Rural GP**	Urban	**Rural**
14	PGY3	Unsure	**GP**	Rural	**Urban**
15	PGY4	Paediatrics	Paediatrics	Urban	Urban
16	PGY3	Paediatrics	**Surgery**	Urban	Urban
17	PGY3	Unsure	Unsure	Rural	**Urban**
18	PGY1	Paediatrics or GP specialising in children	**Unsure**	Rural	Rural
19	PGY1	Surgery or Anaesthetics	Surgery or Anaesthetics	Rural	Rural
20	PGY1	Unsure	Unsure	Unsure	Unsure
21	PGY1	Unsure	**Anaesthetics**	Unsure	**Rural**

EM: Emergency Medicine; ICM: Intensive Care Medicine.

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
