# Peer review of "Extending a Conceptual Framework for Junior Doctors’ Career Decision Making and Rural Careers: Explorers versus Planners and Finding the ‘Right Fit’"

_ijerph, 2020, doi:10.3390/ijerph17041352_

Round 1
Reviewer 1 Report
The introduction is too short to be follow. I cannot see the purpose of this study. I cannot see how this study can be contributed to other countries. The authors continued to use “we”. This is unprofessional in research and academic papers. There are no theoretical framework and theories for supporting. How can the readers understand how this paper be reflected to other studies? The study did not support by any literature and previous studies. The reader had a very hard time to see what the relationships are. The methodology section is very raw. It does not have any discussions about the qualitative research method. Why this methodology is being selected? Why the researchers decided to select this group of people? How can this group of people be selected? No human protections? How can the authors analyse the data information? The 3.1 section should be explained within the Chapter 2. The themes for this result / study were too raw. For an upper-level study, there must be more themes and results. The results were too raw to be understood. For example, where are the sharing and feedback from the participants? The authors said they have conducted several sections of data collection. Why there are no enough supporting? The discussion is cut off. I cannot see how this result can be connected to any previous studies. How can it be reflected to any previous studies? The conclusion is not completed. Too raw and too short. The authors need to outline the limitation of this study.Author Response
Thanks you to the reviewer for the comments and feedback to this study and paper.
Point 1.
The introduction is too short to be follow. I cannot see the purpose of this study. I cannot see how this study can be contributed to other countries. The authors continued to use “we”. This is unprofessional in research and academic papers. There are no theoretical framework and theories for supporting. How can the readers understand how this paper be reflected to other studies? The study did not support by any literature and previous studies. The reader had a very hard time to see what the relationships are. The methodology section is very raw. It does not have any discussions about the qualitative research method. Why this methodology is being selected? Why the researchers decided to select this group of people? How can this group of people be selected? No human protections? How can the authors analyse the data information? The 3.1 section should be explained within the Chapter 2.
Response 1.
The authors note the comments made by all reviewers regarding the brevity in the description of the introduction in terms of providing a detailed description of Pfarrwallers’ theoretical construct and the qualitative methodology of the study. We wish to highlight to the editors that this manuscript is the 2nd paper of a series of two papers that have been submitted for publication in the same edition of the journal. It appears that the reviewers of the 2nd paper did not review the first paper which will have contextualised the objective, methodology and findings of the 2nd paper, more detailed submitted paper and therefore have no context in which to consider the 2nd paper without undue repetition. Consequently in the reviewed resubmission, the authors would like to note to the editors the repetition in the description of the theoretical construct by Pfarrwaller, but more importantly the repetition of the methodology from the first submitted paper. We accept the discretion and decision of the editors to include the repetition in the methodology or reference to the first paper in the same edition as acceptable ways to report the study in this paper. We have considered this reviewer's comments particularly in the context of two submitted and related papers for this edition.
Point 2.
The themes for this result / study were too raw. For an upper-level study, there must be more themes and results. The results were too raw to be understood. For example, where are the sharing and feedback from the participants? The authors said they have conducted several sections of data collection. Why there are no enough supporting? The discussion is cut off. I cannot see how this result can be connected to any previous studies. How can it be reflected to any previous studies? The conclusion is not completed. Too raw and too short. The authors need to outline the limitation of this study.
Response 2.
We believe our themes adequately reflect our data findings.
The discussion and conclusions have been strengthened as suggested
Reviewer 2 Report
An interesting article with considerable potential. However, in my opinion, this material should be further developed before it is accepted for publication.
Below are some comments (technical and substantive), which may be controversial, but it seems to me that it is worth the authors to lean over these suggestions and consider whether it is worth applying them.
Reference numbers in the text should be placed in square brackets [ ].
The aim of the paper could be better articulated in the Introduction.
Line 56: “Recruitment has been described in detail in our first paper.” - The recruitment process of study participants should either be described here again or the “first paper” should be indicated by giving references of this article.
Line 76: Sample selection - why 21 participants took part in the study?
I am inclined to add a description of the process and the criteria for selecting the sample. This will make it easier for the potential reader to understand both the methodical parts and facilitate the analysis of the rest of the text, without having to look for an explanation in another article. But I will not insist on this. I leave it to the authors for consideration.
Lines 79-88: Why was this subsection (3.2. Themes) separated (as part of the Results chapter)? It is rather general/casual and does not contain the study results.
I would suggest expanding the Conclusions to include recommendations on who and how can use the research results. Reading the summary and introduction, I got the impression that the work can be of an utilitarian nature and the results of the research can be used in practice (which would be a great added value). Did I get the wrong impression? Furthermore, the conclusions are too general at the moment.
Author Response
Thanks you to the reviewers for the comments and feedback to this study and paper.
Point 1.
Reference numbers in the text should be placed in square brackets [ ].
Response 1.
We defer to the editors for reference bracketing style preferred by the journal.
Point 2.
The aim of the paper could be better articulated in the Introduction.
Line 56: “Recruitment has been described in detail in our first paper.” - The recruitment process of study participants should either be described here again or the “first paper” should be indicated by giving references of this article.
Line 76: Sample selection - why 21 participants took part in the study?
I am inclined to add a description of the process and the criteria for selecting the sample. This will make it easier for the potential reader to understand both the methodical parts and facilitate the analysis of the rest of the text, without having to look for an explanation in another article. But I will not insist on this. I leave it to the authors for consideration.
Response 2.
The authors note the comments made by all reviewers regarding the brevity in the description of the introduction in terms of providing a detailed description of Pfarrwallers’ theoretical construct and the qualitative methodology of the study. We wish to highlight to the editors that this manuscript is the 2nd paper of a series of two papers that have been submitted for publication in the same edition of the journal. It appears that the reviewers of the 2nd paper did not review the first paper which will have contextualised the objective, methodology and findings of the 2nd paper, more detailed submitted paper and therefore have no context in which to consider the 2nd paper without undue repetition. Consequently in the reviewed resubmission, the authors would like to note to the editors the repetition in the description of the theoretical construct by Pfarrwaller, but more importantly the repetition of the methodology from the first submitted paper. We accept the discretion and decision of the editors to include the repetition in the methodology or reference to the first paper in the same edition as acceptable ways to report the study in this paper.
Point 3.
Lines 79-88: Why was this subsection (3.2. Themes) separated (as part of the Results chapter)? It is rather general/casual and does not contain the study results.
Response 3.
The Themes subsection has been created in order to describe the two main themes of the paper. The subsequent subsections contain detailed explanation of these two themes, forming the results section of the paper.
Point 4.
I would suggest expanding the Conclusions to include recommendations on who and how can use the research results. Reading the summary and introduction, I got the impression that the work can be of an utilitarian nature and the results of the research can be used in practice (which would be a great added value). Did I get the wrong impression? Furthermore, the conclusions are too general at the moment.
Response 4.
The discussion and conclusions have been strengthened as suggested.
Reviewer 3 Report
Thank you for submitting this interesting and well written piece for peer review. I will recommend the article for publication as it is, but I would like to take this opportunity to make a few suggestions.
You refer to the conceptual framework by Pfarrwaller et al. (2017) - it would be useful to include an image as it is hard to visualise the framework (at least for me). It would also be useful to refer to the conceptual framework in the Discussion - are you adding to or refining the framework?
I wouldn't worry about including the point about phenomenology - it is really quite common in social science to refer to participants' lived experience without a nod to phenomenology.
Given that your study is not based on a representative sample of junior doctors, I would be more careful with references to the proportion of participants in certain categories - it may be only 2 people out of 21, but as it is a purely qualitative piece, it does not really matter. However, I agree with including the numbers as a way of describing your sample.
Author Response
Thank you to the reviewer for the comments and feedback to this study and paper.
Point 1.
You refer to the conceptual framework by Pfarrwaller et al. (2017) - it would be useful to include an image as it is hard to visualise the framework (at least for me). It would also be useful to refer to the conceptual framework in the Discussion - are you adding to or refining the framework?
Response 1.
The authors note the comments made by all reviewers regarding the brevity in the description of the introduction in terms of providing a detailed description of Pfarrwallers’ theoretical construct and the qualitative methodology of the study. We wish to highlight to the editors that this manuscript is the 2nd paper of a series of two papers that have been submitted for publication in the same edition of the journal. It appears that the reviewers of the 2nd paper did not review the first paper which will have contextualised the objective, methodology and findings of the 2nd paper, more detailed submitted paper and therefore have no context in which to consider the 2nd paper without undue repetition. Consequently in the reviewed resubmission, the authors would like to note to the editors the repetition in the description of the theoretical construct by Pfarrwaller, but more importantly the repetition of the methodology from the first submitted paper. We accept the discretion and decision of the editors to include the repetition in the methodology or reference to the first paper in the same edition as acceptable ways to report the study in this paper.
Point 2.
I wouldn't worry about including the point about phenomenology - it is really quite common in social science to refer to participants' lived experience without a nod to phenomenology.
Response 2.
We have included the reference to phenomenology for completeness.
Point 3.
Given that your study is not based on a representative sample of junior doctors, I would be more careful with references to the proportion of participants in certain categories - it may be only 2 people out of 21, but as it is a purely qualitative piece, it does not really matter. However, I agree with including the numbers as a way of describing your sample.
Response 3.
We agree that it is helpful for the readers to include a numerical table that describes our sample and have avoided using any statistical analysis given the qualitative nature of this study.
Reviewer 4 Report
as attached.

Author Response
Thank you to the reviewers for the comments and feedback to this study and paper.
Point 1.
(1) This paper has discussed the difference in personalities and its influence on decision making (e.g. exposures, lifestyles and preference) between the “Planners” and “Explorers” groups, which are the most important findings of this study, hence it should be highlighted in both abstract and conclusion.
Response 1.
We have included reference to the "Planners" and "Explorers" in the abstract and conclusion.
Point 2.
(2) Are there any similar findings domestically and internationally?
Response 2.
We have referenced the available literature in this paper.
Point 3.
(3) On page 6 of 9 lines 227-229, Please check “I did my internship at [urban hospital]. I purposefully did not apply for any rural terms because I knew that I wanted to go back rural, and I wanted to get the most city experience that I could in the years that I had to stay in the city. (I05; Female; PGY4)”. Should it be “I did not want to go back rural”?
Response 3.
No, this statement has been quoted correctly.
Point 4.
(4) One conceptual framework, and ecological theory are mentioned in the introduction, while a“four quadrant model” was discussed in the discussion. Could you please introduce all the theoretical models in the introduction to outline this study and then link findings with these models in the discussion?
Response 4.
The four quadrant model is referred to in the discussion as part of a response to the study findings rather than a theoretical model in which the study was testing. It merely forms a brief part of the discussion as an alternate, but disregarded, theory as found in this study’s findings. Consequently it was not included in the introduction.
Point 5.
(5) On page 7 of 9 lines 275-278, it is difficult to understand “Rural background featured prominently in the narratives of doctors who had grown up in the country; however, in our study we found that ‘explorers’ and ‘planners’ were as likely to have a rural background than not.
Response 5.
This statement is clarifying that having a rural background did not appear to influence whether junior doctors fell into the "explorer" category or the "planner" category.
Round 2
Reviewer 1 Report
The author did not respond to the concerns of the reviewer. The paper continued to be unclear to be followed. Although the authors indicated that this is the 2nd series of a study, the reviewer cannot understand any directions of the previous paper. Without any explanations, the reviewers and potential readers cannot understand any background of this study. The authors blamed the reviewer the missing part of the 1st It is horrible that the authors blamed the readers. There are no agreements that potential readers must have the previous background of this study. It is the responsibilities for the authors to indicate the background of the study. The potential readers do not have any responsibilities to have any background of any studies. The reviewer indicated that there are not enough literature reviews and background of the study. The authors did not add ANY additional parts of this paper. The methodology is still very unclear. The reviewer indicated some recommendations. But the authors did not answer my concerns at all. The reviewer indicated that how this study can be contributed to other parts of the world and the communities. However, the authors never responded to this concern. The conclusion is not completed. There are only six lines within the conclusion. The reviewer indicated that this paper needs to have the part of limitations and future research. However, the author ignored the recommendations of the reviewer.Author Response
Thank you to the reviewers, and the editor, for providing feedback and recommendations.
Reviewers 2, 3 and 4 assessed the paper as having appropriate research design, adequate methods, and clearly presented results. Reviewers 1 and 2 recommended further improvements to background, references and conclusions.
The following amendments have been made:
1. References to the first accepted and published paper (for the special edition of the International Journal of Environmental Research and Public Health) by the authors has been included. The abstract has been edited to clarified this is the second paper. Line 12 now reads: “Following our first paper[1], this study….”
2. A paragraph discussing the strengths and limitations of the paper has been included in the Discussion. The discussion includes suggestions as to how this research can contribute to junior doctor training within a conservative approach, based on the findings of the research and within the limitations of the existing literature.
Lines 388 to 398 now read:
“Our study has a number of strengths and limitations. We included a broad range of junior doctors regardless of practice location, specialty intention, and rural exposure. In our first paper [1] we successfully tested one component of Pfarrwaller and colleagues’ model[16] to demonstrate how different systems of influence on career decision-making operate. Furthermore, we explored career decision-making ‘in real time’ rather than retrospectively and identified two phenotypically different groups of students regarding their career decision making. We acknowledge some limitations. Firstly, participants self-selected and, given our recruitment strategy, our sample included an overrepresentation of junior doctors with an interest in rural practice; this, however, does not preclude the validity of our results. Secondly, our study focused on one Australian state with a distinctive population distribution and geographic characteristics, and as a result, some of our findings might have limited applicability in other settings.”
3. Reference style has been formatted as required by the journal.
4. A reference has been included for the John Flynn Scholarship program.
These amendments have improved the paper and we respectfully thank the reviewers, and editor, for their comments.
Reviewer 2 Report
The authors’ answer: “We defer to the editors for reference bracketing style preferred by the journal.”
It is not convincing. I do not understand why the authors did not bother to correct it. It is enough to check the editorial requirements and/or view a few articles of this journal.
Lines 41-42
“We have described the testing of the outer part of the model in our first paper in this Special Edition on Rural and Remote Health.”
Please provide the literature reference of the first article. The reader should know where he/she can read about it.
Author Response
Thank you to the reviewers, and the editor, for providing feedback and recommendations.
Reviewers 2, 3 and 4 assessed the paper as having appropriate research design, adequate methods, and clearly presented results. Reviewers 1 and 2 recommended further improvements to background, references and conclusions.
The following amendments have been made:
1. References to the first accepted and published paper (for the special edition of the International Journal of Environmental Research and Public Health) by the authors has been included. The abstract has been edited to clarified this is the second paper. Line 12 now reads: “Following our first paper[1], this study….”
2. A paragraph discussing the strengths and limitations of the paper has been included in the Discussion. The discussion includes suggestions as to how this research can contribute to junior doctor training within a conservative approach, based on the findings of the research and within the limitations of the existing literature.
Lines 388 to 398 now read:
“Our study has a number of strengths and limitations. We included a broad range of junior doctors regardless of practice location, specialty intention, and rural exposure. In our first paper [1] we successfully tested one component of Pfarrwaller and colleagues’ model[16] to demonstrate how different systems of influence on career decision-making operate. Furthermore, we explored career decision-making ‘in real time’ rather than retrospectively and identified two phenotypically different groups of students regarding their career decision making. We acknowledge some limitations. Firstly, participants self-selected and, given our recruitment strategy, our sample included an overrepresentation of junior doctors with an interest in rural practice; this, however, does not preclude the validity of our results. Secondly, our study focused on one Australian state with a distinctive population distribution and geographic characteristics, and as a result, some of our findings might have limited applicability in other settings.”
3. Reference style has been formatted as required by the journal.
4. A reference has been included for the John Flynn Scholarship program.
These amendments have improved the paper and we respectfully thank the reviewers, and editor, for their comments.